# Home birth preference, childbirth, and newborn care practices in rural Peruvian Amazon

Irene Del Mastro N.●[1]*, Paul J. Tejada-Llacsa[2], Stefan Reinders[2], Raquel Pérez[3], Yliana Solís[2], Isaac Alva[4], Magaly M. Blas[2]

1 Department of Sociology, University of California, Los Angeles, California, United States of America, 2 Epidemiology, STD, HIV Research Unit, School of Public Health and Administration, Universidad Peruana Cayetano Heredia, Lima, Peru, 3 Ages of Life and Education Research Group–EVE, Pontificia Universidad Católica del Perú, Lima, Peru, 4 Intercultural Citizenship and Indigenous Health Unit, School of Public Health and Administration, Universidad Peruana Cayetano Heredia, Lima, Peru

* idelmastro@ucla.edu

**Data Availability Statement:** All relevant data are within the manuscript and its Supporting Information files.

## Abstract

Home birth is very common in the Peruvian Amazon. In rural areas of the Loreto region, home to indigenous populations such as the Kukama-Kukamiria, birth takes place at home constantly. This study aims to understand the preference for home births as well as childbirth and newborn care practices among Kukama-Kukamiria women in rural Loreto. Following a case study approach, sixty semi-structured, face-to-face interviews were conducted with recent mothers who experienced childbirth within one year prior to the interview, female relatives of recent mothers who had a role in childbirth, male relatives of recent mothers, community health workers, and traditional healers. We found that for women from these communities, home birth is a courageous act and an intimate (i.e. members of the community and relatives participate in it) and inexpensive practice in comparison with institutional birth. These preferences are also linked to experiences of mistreatment at health facilities, lack of cultural adaptation of birthing services, and access barriers to them. Preparations for home births included handwashing and cleaning delivery surfaces. After birth, waiting for the godparent to arrive to cut the cord can delay drying of the newborn. Discarding of colostrum, lack of skin-to-skin contact as well as a range of responses regarding immediate breastfeeding and immediate drying of the baby were also found. These findings were used to tailor the educational content of the Mamas del Rio program, where community health workers are trained to identify pregnancy early, perform home visits to pregnant women and newborns, and promote essential newborn care practices in case institutional birth is not desired or feasible. We make recommendations to improve Peru's cultural adaptation of birthing services.

**Funding:** MMB (3) received: Kuskaya: An Interdisciplinary Training Program for Innovation in Global Health Award funded by the Fogarty International Center of the NIH (https://www.fic.nih. gov/); Grand Challenges Canada (GCC) (https:// www.grandchallenges.ca/); and Fondo Nacional de Desarrollo Científico, Tecnológico y de Innovación Tecnológica funded by The Peruvian National Council of Science and Technology (Concytec) (https://www.gob.pe/concytec).

**Competing interests:** The authors have declared that no competing interests exist.

# Introduction

According to Peru's 2018 Demographic and Family Health Survey (ENDES), the most populated region in the Peruvian Amazon, Loreto, has the lowest percentage of institutional births in the country: 74.5% (95% confidence interval (CI) 66.9–80.9)1. This rate is below the national average (92.6%) [1]. In rural areas in Loreto, the percentage of institutional birth is even lower (48.79%, 95% CI: 36.5–61.2). Infections are the main cause of neonatal death in this region, even among newborns with a healthy weight for gestational age and without genetic conditions that might compromise their life [2]. Moreover, between 2010 and 2012, the neonatal mortality rate in Loreto was 18.7 per 1,000 live births, more than double the national rate (8 per 1,000 live births in 2013) [3]. In order to understand the sociocultural factors contributing to these trends, in this paper, we describe the home birth preferences as well as the childbirth and newborn care practices of the of the Kukama-Kukamiria people, an indigenous group in the Peruvian Amazon.

The World Health Organization recommends promoting the use of institutional health services in order to ensure safe childbirth and prevent both maternal and neonatal mortality [4]. During institutional births, trained health professionals with the necessary equipment and supplies can rapidly act if complications occur [4]. In Peru, such institutional births are not feasible for some social groups (e.g. indigenous people) due to women's preferences or their lack of access to health facilities. For example, among some indigenous people, women prefer to give birth at home because it is a sign of courage and prestige [5]. Women who stay at home can access traditional beverages to induce labor such as herbal infusions [6–8]. Home birth also involves the participation of family and community members that help with childbirth and the newborn's physical and spiritual care, while also providing emotional support to the mother [5, 7, 9]. Regarding access barriers, studies have found that indigenous people in the Amazon do not trust health centers due to mistreatment and unnecessary procedures [7, 10]. Moreover, people find health facilities unwelcoming, ineffective, and time-consuming [11]. Little is known, however, about how traditional newborn care practices hinder institutional births or the perspective of community members (e.g. traditional birth attendants, community health workers) about childbirth preferences.

The Peruvian government has issued protocols and health policies aimed at making the country's institutional birthing services culturally sensitive [12–14]. The most thorough of these protocols is the standing birth protocol issued by the Ministry of Health in 2016, which is based on a prior protocol from 2005 [13, 14]. These protocols not only prepare health providers to support a safe birth while respecting the woman's preferred childbirth position, but also help recreate other aspects of home birth [12, 13]. The protocol allows the presence of one companion chosen by the woman giving birth, and it indicates which herbal infusions can be safely consumed by her as well as the supplies needed to assist in a standing birth (e.g. rope, mat) [12, 14]. These protocols do not mention the integration of traditional newborn care practices with institutional birthing services. This protocol applies to every health facility run by the government nationwide except for level one health posts which are too small to offer birthing services [12]. By 2009, most health facilities in the Peruvian Amazon had not incorporated the standing birth protocol from 2005 [15], and there is no evidence that they currently meet the cultural needs of indigenous women.

In a context of slow cultural adaptation of health facilities, community-based interventions can substantially reduce neonatal mortality through the incorporation of clean birth and newborn care practices to home births [16, 17]. In their systematic review and Delphi expert opinion process of clean birth and postnatal care practices, Blencowe and colleagues concluded that clean birth practices such as handwashing and clean delivery surface reduce neonatal

sepsis deaths at home by 15% (IQR 10–20) [16]. Postnatal care practices at home such as clean cutting of the umbilical cord can reduce neonatal sepsis deaths by 40% (IQR 25–50) [16]. According to the authors, clean birth and newborn care interventions should be promoted through community-based behavioral change interventions, training of health workers and birth attendants, and clean birth kits [16]. It is unclear who the specific targets of these strategies are except for health workers and birth attendants. In order to successfully effect both cultural adaptation of institutional birthing services and clean birth and newborn care interventions at home, a broader understanding of home birth preferences as well as childbirth and newborn care practices is required.

Given the high neonatal mortality and home birth rates in rural Peruvian Amazon, the slow cultural adaptation of institutional birthing services, and the efficacy of community-based interventions aimed at promoting both clean birth and newborn care practices, this study aims to understand the preferences for home births and investigate childbirth and newborn care practices among an indigenous group from the Peruvian Amazon. Results were used to develop a community-based intervention that trains community health workers to improve maternal and newborn care practices during home birth and could contribute to the cultural adaptation of institutional birthing services in the Peruvian Amazon.

## Methods

### Study setting

Our study was conducted within the territory of the Kukama-Kukamiria indigenous group community with an estimated population of 69,822 people [18]. The vast majority of the Kukama-Kukamiria people live in the regions of Loreto and Ucayali, which are located in the Peruvian Amazon. We conducted our study in two districts of Loreto. Indigenous communities in this geographic area are widely dispersed; their access to health facilities is time-consuming and by boat only.

The Kukama-Kukamiria people depend mainly on a subsistence economy based on fishing and horticulture. Some of them engage in small-scale commercial activities such as the sale of fine woods and agricultural products (e.g. rice, corn, bananas, cassava, and beans) [19]. More than half of the population lives in poverty in both Nauta and Parinari where people do not have access to electricity, running water, or sanitation [20].

Most of the Kukama-Kukamiria people speak Spanish, but usually the elderly also speak their indigenous language which belongs to the Tupi Guarani language family [18]. There are education policies to teach children from the community the Kukama-Kukamiria language at school [18].

### Design

Our qualitative design was an interview-based case study [21]. This type of study looks for an in-depth description of people's experiences and beliefs [21]. Since we wanted to understand the particular case of home birth practices among the Kukama-Kukamiria people and their preference for home birth over institutional birth, a case-study approach was the best-suited qualitative design for this research.

We conducted face-to-face semi-structured interviews in two phases. During the first phase in May 2017, we explored the general beliefs, customs, and healthcare seeking practices regarding pregnancy and childbirth. The second phase took place in July 2017 with the aim of corroborating findings from the first phase and exploring home birth and newborn care practices in more depth. Interviews were conducted in six villages located in the districts of Nauta and Parinari.

## Target population

Our sample size is 60 participants and was the result of data saturation. The target population consisted of: (1) recent mothers who had experienced childbirth within one year prior to the interview (n = 19), (2) female relatives of recent mothers who had a role in childbirth (n = 12), (3) male relatives of recent mothers (n = 7), (4) community health workers (n = 12), and (5) traditional healers (n = 10). The latter included traditional birth attendants, healers, *vegetalistas*–local experts in the medical use of herbs and vegetables, and *sobadoras*–women who give massages to pregnant women to accommodate the baby in the womb.

## Data collection

A fieldwork team comprised of authors R.P., an expert in qualitative methods, and Y.S., a professional midwife, were in charge of data collection. Before entering the communities for the first time, the fieldwork team sent letters of introduction to the authorities. Once at the community, the fieldwork team approached the community's leaders to introduce themselves and ask for permission to conduct research. In these meetings, the fieldwork team explained the objectives of the study. After gaining the community leaders' approval and support, the fieldwork team presented the research project to the rest of the community and announced that they would invite people to participate in interviews. Our key informant, a community health worker, helped the fieldwork team identify potential participants based on the inclusion criteria. These participants were approached by the team at their houses where they received an explanation of the purpose of the study, the risks and benefits of participation and who to contact in case they had questions or concerns.

Interviews were conducted in Spanish by R.P. who was supported by Y.S. taking notes. This method was designed based on previous research with the targeted population [6–10] and was not pre-tested. All interviews were audio-recorded after written, informed consent was obtained from all participants in accordance with this study's Human Subject Research Protocol (#63631) approved by the Institutional Review Board from Universidad Peruana Cayetano Heredia on April 18, 2017. Interviews took place at the participants' houses, and they lasted between 45 and 60 minutes. The interview guide for the first phase of the study had seven sections: (i) Demographic Data, (ii) Attitudes and Practices during Pregnancy, (iii) Attitudes and Practices during Childbirth, (iv) Attitudes and Practices in Neonatal Care, (v) Local and Traditional Medicine, (vi) Attitudes and Practices in Postnatal and Maternal Care, and (vii) Perceptions about Community Health Workers. Since there were five types of participants, the interview was tailored to each group so that they were asked about their particular roles and perceptions. The data from sections ii, vi, and vii of the interview guide are beyond the scope of this paper so they were excluded from the analysis.

During the second phase of data collection, we used the same interview guides but we spent more time on questions related to traditional practices and newborn care during home birth.

## Data analysis

All audio records were professionally transcribed and pseudonyms were assigned to each participant. We conducted a thematic analysis by following both an inductive and deductive approach since some themes were based on the structure of the interview guide (themes we wanted to explore) and other themes emerged from the data [22]. The first author, I.D.M., read the interview and a random sample of five transcripts. This process lead to the first version of the coding book based on the main themes that emerged from the data and themes that were identified in advance by the research team based on our study goals. After this, authors I.

D.M. and P.T. coded the same two transcripts separately, compared and discussed their codes, and agreed on a new version of the coding book.

Authors I.D.M. and P.T. shared the coding book and the main themes that had emerged with the rest of the research team. In this meeting, the research team collectively decided to organize the themes for this paper into two parts–home birth preferences and childbirth and newborn care practices–and to focus the rest of the coding on those sections of the interview guide. Other sections, such as Attitudes and Practices during Pregnancy, Attitudes and Practices in Postnatal and Maternal Care, and Perceptions about Community Health Workers were excluded from the rest of the analysis process because these topics are beyond the scope of this paper. I.D.M. then coded the rest of the interviews and created an Excel database to serve as an analytic framework consisting of codes in rows and participants in columns with each cell containing quotes. Finally, I.D.M. looked for trends and dissenting positions and analyzed them while also highlighting the most representative quotes of each theme and translating them into English. To ensure translations were accurate, other bilingual members of the research team and a professional English editor revised them.

### Ethics

The Institutional Review Board from Universidad Peruana Cayetano Heredia approved this project´s protocol with code 63631 on April 18, 2017. Written, informed consent was obtained from all participants before interviews.

## Results

### Characteristics of study participants

Over 27 days, we visited six communities and recruited 60 participants. On average, we stayed in each community for 5–6 days. We conducted interviews with 33 participants during the first phase and 27 during the second.

Table 1 shows the demographic information of the study participants. More than half of the study participants were women (39/60). The median age of the participants was 39. One third of the participants (19/60) were recent mothers (RM) who had experienced childbirth within one year prior to the interview. Recent mothers were between 18 and 40 years old and their median age was 27.5. The median number of children among recent mothers was 4.

Table 1. Demographic information of study participants.

| Charactersitics of Participants | Recent Mothers (n = 19) | Female Relatives (n = 12) | Male Relatives (n = 7) | Community Health Workers (n = 12) | Traditional Healers (n = 10) |
|---|---|---|---|---|---|
| **Sex** | | | | | |
| Female | 19 | 12 | n.a. | 5 | 5 |
| Men | n.a. | n.a. | 7 | 7 | 5 |
| **Age** | | | | | |
| Lowest value | 18 | 18 | 24 | 18 | 37 |
| Highest value | 40 | 74 | 44 | 58 | 71 |
| Median | 27.5 | 51.5 | 34 | 49.5 | 46 |
| **Number of children** | | | | | |
| Lowest value | 1 | 0 | 1 | 0 | 2 |
| Highest value | 10 | 12 | 13 | 10 | 14 |
| Median | 4 | 8 | 4 | 2 | 6 |

The female relatives group (FR) was diverse in age since it was comprised of mostly sisters, mothers, and grandmothers of the recent mothers. The youngest female relative was 18 years old and the oldest was 74. The median number of children for female relatives was 8. A total of seven male relatives (MR) of recent mothers were interviewed. They were either their fathers or partners and their median age was 34. Ten participants were traditional healers (TH) and five of them were older traditional birth attendants. We also interviewed 12 community health workers (CHW), five women and seven men, whose median age was 49.5 years old. Community health workers are trained to monitor new pregnancies. Their role is to conduct home visits to pregnant women and newborns to check on any signs of health complications and report them to the health facility.

## Home birth preferences

Study participants reported that most women in their communities prefer to give birth at home instead of a health facility. Feeling safe and supported are the main reasons to stay at home for childbirth. This is contrasted with a fear of mistreatment and inadequate care at health centers. Other related factors include home birth being perceived as brave and partners' and social expectations that support home birth. Access barriers such as financial and transportation constraints and long wait times are other complementary factors that limit the use of birthing services.

**Safety and support vs. mistreatment and inadequate care.** Most participants from all groups indicated that home birth is a traditional practice in which women feel safe and supported by relatives and traditional healers. On the contrary, institutional birth is associated with mistreatment and inadequate care, which was emphasized mostly by community health workers, recent mothers, and female relatives. During home birth, relatives and traditional healers provide emotional care and practical support through food, drinks, and traditional medicine. María, a 23-year-old recent mother, shared her preference for home birth after having three children at home:

> I [prefer to give birth] at my house. I feel safer here than at the health center because people say that at the health center, you are not treated like when you are at your house. They do not let you take [traditional] medicine, those that you take here [at home]. They [traditional birth attendants and relatives] give you a lot of hot beverages, and there [at the health center] they do not, people say. And if you cannot have your baby, if your baby does not dilate well, they take you to the city. Here [at home], it is not like that; here they [traditional birth attendants and relatives] see how to make it work, they make your baby dilate fast with vegetables.

Relatives of recent mothers also expressed their preference for home births. Being able to take care of the relative giving birth was the main reason reported:

Interviewer (I): How do you feel about your daughter giving birth at home?

Participant (P): Oh, I feel calm because I see her next to me, I am looking after her, I am taking care of her. She asks me for something and I give it to her. Maybe [she asks] for a stew that maybe she wants to drink. Maybe she asks me: "Mom, I want to drink" and I give it to her, I boil it for her. I take care of her here.

While women feel safe and supported having a home birth, community health workers, recent mothers, and their female relatives shared secondhand and personal stories of women

being told off for not pushing during labor and receiving unnecessary procedures such as C-sections:

> Here [at home] I am motivated by my father-in-law and my mom when they tell me: "Daughter, walk, walk, have courage, if you do not have courage, who is going to have it for you? Nobody." At the health center, you are told off; they are yelling at you.
>
> *- Recent mother*

> Well, some [women] say they do not want to go to the hospital because at the hospital, they perform C-sections on them, they cut them. Because of that, they are afraid to go to the hospital; they prefer to give birth in their houses with the traditional midwife instead of getting cut. That is what I hear many women who have children saying because that is the fear they have, to be cut.
>
> *- Community health worker*

> [Women don't go to the health center] because they [health providers] grope them, they do things like that to them so it is better to be at home. Sometimes, they [health providers] give them [women in labor] cuts, cuts so they dilate faster so they have their kids more quickly.
>
> *- Female relative*

Along with perceived mistreatment, many interviewees were concerned about inadequate care at the health centers. The woman being left alone during labor or forced to wait to receive care were salient themes when participants described why women prefer to give birth at home. Participants from all groups reported these themes. For example, a male relative explaining why his wife does not trust the health center said, "She is suspicious that they might leave her there alone [during labor] and she could die. That is where her mistrust comes from. [At the health center], they leave them all alone." A recent mother emphasized the long wait saying that the health providers "take us to the health center, then the [nurse] technician leaves us there alone, and they treat us only when we can't walk anymore."

Participants' perceptions of inadequate care also stemmed from a series of restrictions to their traditional childbirth practices at the health center such as having a relative or community member cut the umbilical cord, the consumption of traditional beverages, and having a standing–or any position except horizontal–birth. These restrictions contribute to women's preference for home birth.

Participants from all groups reported that during home birth, women drink hot beverages that are not provided or allowed at health centers such as cotton leaf tea, milk, coffee, and *ubos*–a slightly alcoholic beverage made out of fermented hog plum. For instance, a participant said, "[Health providers] do not let you take your medicine, those that we take at home."

Some recent mothers and their female relatives shared that they decided not to give birth at the health center because if they did, their babies could not have a godmother. It is a common practice among Kukamas that a godparent chosen by the parents cuts the umbilical cord. Silvia, a 29-year-old recent mother, explained why she did not go to the health center to give birth: "because I wanted her [my baby] to have a godmother, because at the hospital they do not want [the godmother to cut the umbilical cord], it is not possible." A father of two children born in health centers who did not have their cords cut by a godparent said that he feels "kind of weird because you should have a godfather at birth."

Although standing birth is starting to be offered at some health centers across the country, many women are not aware of this or do not live close to a center that offers this option. The

lack of this service discourages women from having an institutional birth. For example, Flavia, a 21-year-old woman who has had three pregnancies, says that she and other women from her community prefer to give birth at their houses because: "Here [at home] you have [birth] in your normal way, they [health providers] are not touching you. And here [at home] you have [birth] in the position you want, there you don't."

**Brave women.**   Most participants from all groups reported that women who have home births are brave and have a lot of courage whereas women who go to health centers are weak or "dejadas." For example, a father of six said that his partner's last birth was at home because "she had courage." Romina, a grandmother and a mother of eight, shared what she was told by her closest female relatives when she was pregnant the first time:

> Back in the time when my grandma was alive, she would tell us that we [had to] give birth there [at home]. If we were brave, we could give birth in our house; if we were not, we would be taken to the health center, she said.

A traditional healer also emphasized the contrast between having a home birth and being brave and having an institutional birth and being weak:

> Not everyone can give birth. Especially the first time. Many of them scream: "Ay, ay." They talk nonsense. Those things sometimes intimidate the husband, it intimidates the family, the mother, the grandmother. Then, they take her [the pregnant woman] to the health center.

The perception of women who go to health centers as weak or "dejadas," however, was contested by a few participants from each group. These participants considered that during emergencies, giving birth at the health center is not a sign of weakness. For some of them, women giving birth are always brave regardless of where it happens: "Both of them are brave, those [who give birth] at the health post and those [who give birth] at home. Women are always brave when they have to bring a child into the world" (male relative, father of 13 children).

**Gendered factors.**   Regardless of their group, some women from the study reported having a home birth because it was their partner's/husband's decision. Nadia, an 18-year-old recent mother, said that she wanted to give birth to her child at home because "that is what my husband said." A similar dynamic was described by a community health worker when she was asked who decided the place for the birth: "The pregnant woman decides and her role is to tell her husband so he can accept or not because some [husbands] do not accept."

Women's preference for home birth is also informed by gender when the health provider in charge of the woman's health is a man. Participants, most of them recent mothers and community health workers, reported that women feel ashamed showing their bodies to male health providers. When Martín, a community health worker with 22 years of experience was asked why women do not go to the health center during pregnancy or labor, he said, "I mean, for one, the [professional] midwife is a man. They are embarrassed by that."

**Access barriers to health centers.**   Besides the cultural and interpersonal aspects of home birth, participants from all groups mentioned features of health centers that contribute to preference for home birth. Long distance, cost, and waiting time were main reasons for not having an institutional birth.

Community health worker Martin explains the economic reasoning:

> I mean, I think that [they do not have institutional births] because of their finances. Okay, I am going to tell you, sometimes there is no money, not even for a gallon of gasoline. Mainly

that, that is why mothers are used to giving [birth] at home. They do not want a hospital nor a health center. They [women] are used to giving birth at home.

Julia, whose daughter recently gave birth, explains that distance is another reason she prefers giving birth at home:

Of course [it is easier to give birth at home]. If I go to the health center and I am in a hurry, for example, maybe I cannot get a towel or a diaper for the baby. Look how far away I live [from the health center]. It would take too long to come here to pick up something from the health center. If it is at home, it is easier, and I do not have to move around with my daughter.

In addition to distance and transportation barriers to health centers, once they are there, the people in these communities also have to deal with long waits to receive care. Marlene, a 22-year-old mother of three, shared why she decided to have all her kids at home:

I like it [giving birth at home] because, well, at the hospital, they take too long to take care of you, they ask you for so many things [identification, paperwork, to wait and give birth without relatives in the room]; they delay you actually. At my house, well, that is where I had [my children]. There [at home], my mom, they give you everything hot, they give you, so you drink and you have [your children] fast. That is how I had all my children.

## Birth and newborn care practices

After exploring the main reasons why most women in these communities prefer to have home births, we analyzed how these births happen. We focus on aspects directly related to newborn health such as clean birth procedures and newborn care.

**Safe birth: Handwashing and clean surfaces.** Regarding safe childbirth practices, all participants who were asked about handwashing said that the traditional midwife or relatives attending the home birth do wash their hands with soap before labor. This information came from traditional birth attendants reporting about their own practices as well as community health workers and recent mothers and their female relatives.

Despite the male relatives group who reported not knowing about the cleaning practices before childbirth, all participants mentioned that babies are delivered to a blanket on a previously swept wood floor. The blanket is washed either with rain or river water before the birth takes place. There was only one participant, a recent mother, who did not know if the blanket used during her labor was washed before or not because her mom was in charge of that. A piece of plastic was used by a few participants which goes under the blanket or it replaces it. The next excerpt represents these findings:

P (recent mother): We had put the mattress in one side of the room and the plastic. I had plastic and I put some blankets. It was all clean. Always the room [has to be] clean. It was like this, first plastic, right? Then my sheets so they could get dirty.

I: How were the sheets washed?

P: They are washed with rain water.

**Newborn care: Immediate drying and breastfeeding, skin-to-skin contact, and cutting the umbilical cord.** After the baby is born, it is not always dried immediately. Although most of our participants who commented on this said that the baby is dried immediately after being

born, a few community health workers revealed that the newborn can wait up to 30 minutes to be dried. This usually happens because the baby is placed on a blanket until the godparent arrives to cut the cord: "[The baby waits] like 30 minutes, until we find the godfather" (community health worker). It is not until that procedure is done that the baby is dried.

We found that scissors were the tool used to cut the cord according to the vast majority of participants from all groups. These scissors were either bought specifically for the birth, previously owned or, in fewer cases, brought by a health provider. In any case, they were always disinfected with boiling water.

Another key aspect of newborn care is breastfeeding. Although responses regarding whether or not breastfeeding was immediate–within one hour of birth–varied, the majority of participants agreed that the first breastfeeding took place after the baby was cleaned and dressed. According to our participants, this can occur between 15 to 120 minutes after the baby is born. In one extreme case, breastfeeding occurred 13 hours after birth because the baby "did not ask for it" according to the mother.

Regarding skin-to-skin contact between mother and baby, according to most participants, this first direct contact is not immediate since it happens once the baby is cleaned by the traditional midwife or the relative helping with childbirth. For instance, Vilma, a 53-year-old mother of eight, said that her daughter carried her baby "after the traditional midwife cleaned him, that is when she gives it to the mom." When this contact occurs, the baby is already dressed. For example, a community health worker, clearly stated that the baby's first contact with the mother was after she was dressed: "The baby was in the blanket, then we cleaned and dressed her, and once she was well covered, the mother carried her." There was only one participant, a traditional birth attendant, who reported skin-to-skin contact when she explained the procedures after birth. According to her, the baby is placed naked on the mother's naked belly until the godparent arrives to cut the cord.

As for the colostrum, our results show that discarding it is a common practice. Whereas male relatives did not show much understanding about the colostrum, the majority of female participants mentioned that the first breast milk, the colostrum, is discarded. This happens because people think that the colostrum makes the baby sick because it is old and stored milk. Laura, a 74-year-old mother of 11, explained that she discarded the colostrum because "the baby gets infections in his stomach." Sofia, a traditional midwife with 20 years of experience said that she "throws out the yellow milk [colostrum]" because it makes "the baby throw up."

Although the majority of participants reported the practice of disposing of the colostrum, a few of them keep it and give it to the newborn. Rafaella, a mother of four who gave birth eight days before the interview, shares how she breastfed all of her children without discarding the colostrum:

I. Did you wring out that milk [colostrum] and throw it away? Or did you give it to the baby?

P: No, I gave it to them directly, they [the professional midwife] did not explain anything to me. They told me to breastfeed him and I did.

I: How was it with your other children?

P: The same.

I: You have never wrung out the yellow milk.

P: Never.

## Discussion

This study explores the Kukama-Kukamiria people's preference for home births and their childbirth and newborn care practices. Our findings show that women prefer home births over institutional births because it is a culturally charged practice that they cannot reproduce at a health center. Home birth makes women feel safe and supported. It is also a sign of bravery and a way to comply with traditional gender roles such as following their partner's decisions. These preferences are also linked to experiences of mistreatment at health centers, lack of cultural adaptation of birthing services, and access barriers. Regarding newborn care, there are practices that contribute to hygienic childbirth such as handwashing and clean delivery surfaces. However, the discarding of the colostrum, the lack of skin-to-skin contact, and the range of responses regarding immediate breastfeeding and immediate drying are findings that show opportunities to improve the newborn care practices in these communities.

We found barriers to both health facility access and quality care. In relation to access, studies in the Peruvian Amazon and other low and middle-income countries have also found that distance and financial constraints limit people's options to home birth [10, 23–26]. When exploring the causes of maternal death in Peruvian rural areas, Anderson found that money, distance, and transportation are factors that delay the provision of care for pregnant women, which can put them at risk of maternal death [23]. The evidence in Nepal suggests that other barriers to skilled birth care include the limited availability of skilled birth attendants and lack of community awareness regarding the importance of birthing services [27]. As for healthcare provision, a study among the Kukama-Kukamiria people found that going to a health center was considered a last resort because of their experiences of mistreatment [15]. Moreover, Avellaneda reported health providers' lack of awareness of Kukama-Kukamiria traditional health practices and their unwillingness to incorporate such practices [6].

In accordance to this study's findings, Avellaneda, Rivas, Otto, and Leon have also reported the use of traditional infusions and hot beverages to induce labor such as cotton leaf tea and *malva* or mallow, ginger, coffee, hot milk, tea, and hot water among Kukama-Kukamiria, Yaguas, and Awajun people [6–8]. Rivas also reported the use of black pepper, cocoa, coca leaf infusions, cooking oil, egg white, mallow leaf juice, and paracetamol [6, 7]. Our study shows that the lack of incorporation of those practices make women less interested in having an institutional birth. Health providers should follow the national protocols on standing birth, which also allows the use of certain traditional beverages [14].

Some women from this study reported feeling unsafe and uncomfortable at the health facility, especially if the provider was a man. In their study about women's home birth preferences in rural Bangladesh [25], Saker and colleagues found that the lack of female doctors in the health facilities also contributed to this preference. Feelings of discomfort and unsafety, however, do not come exclusively from being treated by a male practitioner. They are also related to the lack of appropriate care. An ethnographic study in a Kukama-Kukamiria community found that despite health providers speaking Spanish as the majority of Kukama-Kukamiria people do, the health center was perceived as an unwelcoming place due to mistreatment and the lack of timely and effective care [7, 11].

Mistreatment in the health center such as receiving unconsented intrusive procedures is an important barrier to institutional births. A cross-sectional study in four low-income and middle-income countries found that 75% of the observed women who received an episiotomy did not consent to one [28]. Moreover, in 13% of the childbirths observed, women received a caesarean section without consenting to one [28]. In her study on the causes of maternal death in rural areas of Peru, Anderson found that women and their partners do not go to the health center to avoid health providers making "internal cuts" to them [7, 23]. A study conducted in

14 regional hospitals located in nine Peruvian cities, including two cities in the Amazon, found that violence during childbirth was reported by 97.4% of women (n = 1488) [29]. These findings reinforce the distrust that indigenous people have towards health services caused by the forced sterilization campaign conducted by the Peruvian government in the nineties [30]. Peruvian providers' perspectives of mistreatment during institutional births is unknown, but studies in other countries suggest that providers are aware of verbally and physically abusing women themselves and observing their peers doing the same [31]. Some of the drivers of mistreatment and abuse are perceiving women as being difficult, lack of accountability, and stress [31]. Studies in other Latin American countries with a high proportion of indigenous people show that experiencing mistreatment during institutional births lowers the probability of returning to the health facility [32].

While institutional births are associated with access barriers and poor quality of care, our study shows that home birth represents a cultural practice that makes women feel safe and supported. Studies show that this satisfaction with home births leads women to maintain their preference for home births [32]. Other studies in low and middle-income countries have also shown that women prefer home births because it is a cultural practice that provides comfort, intimacy, and autonomy to women [33] while complying with their communities' traditional views of childbirth [25]. Studies about home birth preferences in Brazil and Zimbabwe have found that home birth entails the participation of family and community members that provide emotional support to the mother and help with childbirth and the newborn's physical and spiritual care [26, 34]. Qualitative studies of the Kukama-Kukamiria people show that the presence of relatives during childbirth is understood to drive away bad spirits [5]. The baby's father also has an important role during childbirth because he motivates the mother when she is pushing [7, 35, 36]. According to the current standing birth protocol in Peru [14], pregnant women can choose one companion during institutional birth. This can be the father or any other family or community member. In light of our findings and related literature, this norm is insufficient. To make institutional birth more appealing to Kukama-Kukamiria women, more companions should be allowed in the childbirth room.

One key aspect of home births is the role of the godparents. Our findings show that having a godparent cut the umbilical cord is an important reason for having a home birth. The godparent's cultural value has been reported in other studies of the Kukama-Kukamiria people [9]. It has also been reported that it is the godmother that makes the newborn become a human being [5]. The Kukama-Kukamiria people consider the relationship between the baby and the godmother the most important spiritual relationship that a person can have [7]. Our study contributes to literature on the cultural meaning of the umbilical cord cut by adding a new case that demonstrates the importance of the godmother in this process. It also contributes to it by showing that not integrating the role of the godparents into birthing services can be a barrier to institutional birth. Additionally, we found that the cut of the umbilical cord can be delayed if the godparent is not present during childbirth.

Another cultural aspect of home birth that contributes to women's home birth preference is being considered brave. The relationship between home birth, courage, and prestige has been found in other studies among the Kukama-Kukamiria people. These studies have explained that for the Kukama-Kukamiria people, the act of pushing during birth in an intimate space surrounded by relatives is highly valued because women can show their strength. Childbirth is related to strength, which is perceived as the ability to perform a laborious task that requires discipline and implies sweating [37]. This is a demonstration of spiritual vitality. Among Yine women, another indigenous group from the Peruvian Amazon, the socially accepted way to have babies is through physical effort that causes pain and sweat and without any help [37]. Our study not only reinforces these findings but also shows how the association between home

birth and bravery can be a barrier to institutional birth. However, the few people we found who also consider women having institutional births to be brave, suggests a potential cultural change that could lead to more institutional births. To reinforce this cultural change and given what we know about the link between home birth and bravery and the perception of the health center as a place where women are powerless (i.e. subjects of mistreatment and unconsented procedures), we suggest that interventions should emphasize that institutional births are for all women, not just those experiencing complications, and that women can decide their childbirth positions during institutional births. We also recommend allowing more companions than the one that is currently permitted during institutional births.

Although our findings show the positive impact of family and community members in women's preference for home birth (i.e. feeling safe and supported and being considered brave), we also found that home births are related to women's limited decision-making power. In Saker and colleagues' study [25], the decision to have a home birth was made by the men and elderly members of the family. A systematic review of access barriers to obstetric care at health facilities and a literature review of determinants of institutional births show that women's decisions are heavily influenced by their partners' preferences [38, 39]. According to these studies, women need their husbands' or relatives' permission to have an institutional birth, and they may lack control over the material resources necessary to afford or access one [38, 39]. Enhancing women's decision-making capacity has to be a primary goal of any intervention promoting clean home births or institutional births.

In relation to newborn care practices, we detected delays in and lack of skin-to-skin contact and immediate breastfeeding. According to Moore and colleagues, this can be harmful to the newborn and the mother because skin-to-skin contact improves breastfeeding, which is a protective factor against illness for newborns [40]. Other reported advantages of skin-to-skin contact are improving suckling competence, maternal satisfaction, and temperature control of the newborn as well as reducing primary postpartum hemorrhage [39, 41, 42]. Previous studies of the Kukama-Kukamiria people have also found delays in immediate breastfeeding [15, 17, 29]. The lack of skin-to-skin contact that we found should be addressed by health officers. We recommend educational campaigns targeting community health workers, pregnant women, and traditional birth attendants about the importance of skin-to-skin contact. Moreover, we recommend addressing one of the main reasons for the delay of such contact–waiting for the godparent to cut the cord. For instance, recommending that the newborn rest on the mother's chest even if the cord has not yet been cut, is a possibility.

For most mothers in our study, the colostrum is stored milk that hurts the baby and should be thrown away. Kukama-Kukamiria participants from other studies were not familiar with the term colostrum either [9]. They called it "first milk" instead and discard it because they did not know its benefits for the baby [9]. The unique nutritional value of colostrum varies between feeds and stages of lactation [43], but it is known to improve intestinal maturation and has antibodies against multiple enteric pathogens [44, 45]. Further, the WHO recommends immediate breastfeeding within the first hour because it reduces neonatal mortality [46]. Our findings show the negative perception of the colostrum among Kukama-Kukamiria women, which can be targeted by future interventions aimed at improving newborn health.

In relation to newborn care, participants mentioned some beneficial practices such as disinfecting the scissors used to cut the umbilical cord and handwashing. These practices have been reported in other Latin American countries, showing that traditional birth attendants are aware of internationally recommended newborn care protocols [47]. Educational campaigns for newborn care should keep emphasizing the benefits of these practices since there are other studies of the Kukama-Kukamiria people that found people cutting the cord with non-sterilized sharp objects (e.g. razor, knife) that can generate infections [11]. Handwashing in

particular should continue to be addressed at both the community and the government level since access to clean water is limited in this area. Even if participants report washing their hands and childbirth surfaces, the quality of the water used could still put the newborn's health at risk. Clean water and sanitation are paramount for newborn care.

Historically, cultural aspects of childbirth have been considered by the Peruvian government as barriers to safe pregnancy and childbirth since they usually lead to home birth instead of institutional birth. However, several policies aimed at integrating traditional practices into institutional birthing services have been approved since 2005 [12, 13]. For instance, the 2005 standing birth protocol and its updated version in 2016 allow the consumption of certain traditional beverages during labor and the participation of one companion during childbirth [12, 13]. This is important progress, but our study shows that women are not aware of these changes. Moreover, the implementation of the protocol is limited [15]. In our complementary formative research for the Mamas del Rio program, we found that none of the 10 health facilities serving Kukama-Kukamiria people had the 13 supplies (e.g. mats, ropes, stool) required to adapt their services to accommodate the standing birth protocol. Five of those facilities had only two supplies and the other five had none. Since the lack of implementation of such protocols is a barrier to institutional births, a nationwide evaluation of protocol implementation should be conducted. We recommend that the supplies be distributed to every health center, prioritizing those that primarily serve indigenous people, and that communities be informed about the cultural adaptation of birthing services.

This study shows the limitations of the current standing birth protocol and can inform future changes to ensure the cultural adaptation of birthing services in the Peruvian Amazon. For instance, allowing only one companion might be a barrier to institutional births since women are used to having their close relatives and a birth attendant during childbirth. We recommend increasing the number of companions to at least four people so that the partner, the mother, a childbirth attendant, and a godparent can be present. The cultural adaptation of birthing services should also integrate the role of the godparent in cutting the umbilical cord. The mechanisms to report and hold health providers accountable for mistreatment should also be improved and including community health workers in this process is key.

Finally, recognizing that institutional births are not always feasible or desired and focusing on making home births clean and safe is crucial to reduce maternal and neonatal death [16, 17]. Our study shows that hygienic birth practices (e.g. handwashing, scissors sanitization) are common in the community, but structural factors such as access to water and sanitation are necessary to ensure their effectiveness. Based on our study results, home births in the Peruvian Amazon could be safer if practices such as immediate drying, skin-to-skin contact, and breastfeeding including the colostrum are implemented at the community level. To be successfully implemented, however, this biomedical information needs to fit into an existing sociocultural system of childbirth and newborn care instead of directly conflicting with people's customs and experiences [48]. The results of this study are part of the formative research used to build the Mamas del Rio program, which includes a training program for community health workers aimed at improving essential newborn care practices and neonatal healthcare. This is achieved through educational home visits by community health workers to pregnant women and mothers with newborns. Supportive program components of Mamas del Rio include the training of traditional birth attendants, the improvement of healthcare facilities, and comprehensive supervision of community health workers' tasks. We recommend replicating the Mamas del Rio program nationwide.

## Strengths and limitations

This study has a few limitations. Interview-based studies face issues of social desirability and recall bias that can be particularly salient when reporting about healthcare practices that involve vulnerable groups such as newborns. We recommend that future research complements interview data with ethnographic observations of home births and institutional births since this method provides a better understanding of the meaning of practices for participants and gives the researchers first-hand information.

Another limitation of this study is that participants hold multiple roles in their communities but were placed in one specific group. For instance, some female relatives were also traditional birth attendants. As a result, we encountered few differences between groups. This is a challenge when conducting research with small, isolated, and underserved communities where division of labor is limited and social cohesion is strong. This makes people's beliefs and customs very similar, which makes the incorporation of community members in intervention programs crucial to their success.

Qualitative studies like ours have small samples so the results are not representative of the whole country or every indigenous group in Peru. However, these type of studies are rich in description and reveal the mechanisms that lead to certain behaviors like home birth and newborn care. Another strength of our study comes from the two phases of data collection. This gave us the opportunity to follow up and ask additional questions to better understand home birth preferences and childbirth and newborn care practices.

## Conclusion

Our study contributes to the understanding of women's preference for home births as well as childbirth and newborn care practices in the Peruvian Amazon. The Kukama-Kukamiria people prefer home birth over institutional birth because they see it as a meaningful, inexpensive, practical, and socially prestigious practice. On the other hand, institutional births are costly, difficult to access, and put women at risk of mistreatment and abuse. Favorable childbirth practices in home birth are handwashing and cleaning delivery surfaces. The discarding of the colostrum, lack of skin-to-skin contact, and delays in immediate drying and breastfeeding should be targeted by public health interventions to improve maternal and neonatal health outcomes.

## Acknowledgments

We want to thank all of the community members who participated in the study.

## Author Contributions

**Conceptualization:** Stefan Reinders, Raquel Pérez, Yliana Solís, Isaac Alva, Magaly M. Blas.

**Data curation:** Irene Del Mastro N.

**Formal analysis:** Irene Del Mastro N., Paul J. Tejada-Llacsa.

**Funding acquisition:** Magaly M. Blas.

**Investigation:** Raquel Pérez, Yliana Solís.

**Methodology:** Stefan Reinders, Raquel Pérez, Isaac Alva, Magaly M. Blas.

**Project administration:** Stefan Reinders, Raquel Pérez, Yliana Solís, Magaly M. Blas.

**Resources:** Stefan Reinders, Raquel Pérez.

**Supervision:** Magaly M. Blas.

**Validation:** Magaly M. Blas.

**Writing – original draft:** Irene Del Mastro N., Paul J. Tejada-Llacsa.

**Writing – review & editing:** Irene Del Mastro N., Paul J. Tejada-Llacsa, Stefan Reinders, Isaac Alva, Magaly M. Blas.

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
