## [Decision Letter · Decision Letter 0]

26 Oct 2020

PONE-D-20-27331

Home birth preference, childbirth, and newborn care practices in Kukama-Kukamira communities in rural Amazonas

PLOS ONE

Dear Dr. Del Mastro Naccarato,

Thank you for submitting your manuscript to PLOS ONE. After careful consideration, we feel that it has merit but does not fully meet PLOS ONE’s publication criteria as it currently stands. Therefore, we invite you to submit a revised version of the manuscript that addresses the points raised during the review process.

We look forward to receiving your revised manuscript.

Kind regards,

Emma Sacks

Academic Editor

PLOS ONE

Additional Editor Comments:

This is an important study from an under-researched part of the world. However, the paper needs significant revisions. The reviewers have given very thorough and helpful suggestions.

It would help if the introduction included answers to the following: What is new? Is this an update from a previous study? Is there currently a growing stigma for home births? What do people understand about the risk of maternal mortality?

Overall, the manuscript needs significant grammatical editing before it can be considered for publication.

Line 29 – is “every second” literal?

Line 40- how are drying and cord cutting related here? Again on line 443, the question arises again. Delayed cord clamping is actually recommended to increase iron to the baby. Couldn’t the cord be dried even if there was a delay in having it cut?

Line 41- what are “heterogenous” answers?

Line 55 – is this a rate or a percentage?

Line 74- more details needed on this integration. When was this? How many facilities?

Line 102 – accessible to where?

Line 113- were observational data included?

Line 126- please include numbers of respondents and justification for the sample size

Line 130 – spelling should be midwives (or can keep as parteras?); are sobedoras different than TBAs?

Line 141- please include consent process details

Line 143- please include the languages, and if tools were pre-tested

Line 180- why was the stay in each community so short? This would be difficult for a phenomenological study

Line 192- what are the roles and training of CHWs?

Line 222- were experiences of mistreatment expressed directly or second hand stories?

Line 266- please spell out SJS

Line 279- is it only gendered factors, or also intrafamiliar, and other power dynamics?

Line 281- what is “L”?

Line 313- what types of “things” are asked for?

Line 329- are TBAs supposed to get gloves from health facilities?

Line 349- do godmothers and godfathers have different roles?

Line 351- were responses not clear, or just given in a range?

Line 362- is “carried her” the same as skin to skin care?

Line 374- who is “they”?

Line 394- “people want to” should be changed to indicate less choice, given constraints

Line 400- lack of awareness by who?

Line 410- who is “their”?

Line 416- please see the following publication, showing a large percent of women experience unconsented vaginal exams and other disrespectful care - https://pubmed.ncbi.nlm.nih.gov/31604660/

Line 423- what language is spoken at the health care center?

Line 449- this is not clear – what is the link between pushing and having relatives present? (are the fathers of the babies allowed in health facilities?)

Line 465- are other liquids given when breastfeeding is delayed?

Line 466- who is this reported by?

Line 473- please see recent evidence related to the risk of bed sharing - https://pubmed.ncbi.nlm.nih.gov/27940805/

Line 479- the average number of days when colostrum is available is known

Line 484 – is this maternal or neonatal mortality?

Line 486- use word like “beneficial” rather than suitable, which is more judgmental

Line 500 – please provide some examples of cultural adaptations

The discussion would be strengthened with more references to the other work that has been conducted on maternal and postnatal care and careseeking in latin America. Please see the following citations:

https://pubmed.ncbi.nlm.nih.gov/29325572/

https://pubmed.ncbi.nlm.nih.gov/23360944/

https://pubmed.ncbi.nlm.nih.gov/16225975/

If available, please include a demographic table or description of the participants.

Please append the interview guides.

It would be helpful, in the results, to understand the weight of various factors that participants describe that help them decide when and where to seek care.

Please include a strengths and limitations section.

Journal Requirements:

2. Please include a copy of the interview guide used in the study, in both the original language and English, as Supporting Information, or include a citation if it has been published previously.

Reviewers' comments:

Reviewer's Responses to Questions

**Comments to the Author**

1. Is the manuscript technically sound, and do the data support the conclusions?

Reviewer #1: Yes

Reviewer #2: Yes

2. Has the statistical analysis been performed appropriately and rigorously? 

Reviewer #1: N/A

Reviewer #2: Yes

3. Have the authors made all data underlying the findings in their manuscript fully available?

Reviewer #1: Yes

Reviewer #2: Yes

4. Is the manuscript presented in an intelligible fashion and written in standard English?

Reviewer #1: No

Reviewer #2: Yes

5. Review Comments to the Author

Reviewer #1: 1. Summary of the research

This study presents a compelling qualitative analysis of home birth preferences and practices among the Kukama-Kukamira people of rural Amazonas. These findings have significant implications for maternal and infant health as they elucidate the cultural meaning ascribed to home birth, the rationale for its preference over institutional delivery, and homebirth and newborn practices among indigenous communities in the Amazonas. The authors propose that these findings can be used to promote evidence-based maternal and newborn care practices during home birth and inform the integration of cultural practices in institutional birthing facilities.

2. Examples and Evidence

a. Major Issues

It is unclear whether the authors are ultimately recommending improving home birth practices (as is intended with the community-based intervention) or integrating cultural practices into hospital systems to increase institutional births. Even so, health systems must first address the structural barriers impacting access to institutional deliveries, namely distance from facility and cost of transportation. The authors should more clearly indicate what the ultimate implications of their findings are.

The phenomenological and case study approach is appropriate to address the proposed research questions and the theoretical framework was adequately applied. The process of conducting a second round of interviews ensures deeper exploration of the processes being examined.

Authors should state whether the interviews were conducted in Spanish and if so, how they were translated.

I am concerned about how conducting the interviews in participants’ homes may have impacted responses, especially in relation to discussions of partners’ decision-making and gender roles in relation to institutional deliveries as participants may have felt uneasy with speaking openly if their partners were present or nearby.

Please elaborate on the data analysis methodology – were all transcripts (besides the first two that were double-coded) coded by I.D.M and P.T.? What do the authors mean by “we collectively decided how to organize the themes of this paper” in line 168?

The results are comprehensive and the quotes illustrative of the identified themes. One way to improve this section would be to tease apart discrepancies between ideas from different types of respondents – what were the differences in perspectives from mothers, partners, traditional healers, community health workers, and female relatives. For example, did postpartum women mention cost or distance as deterrents to institutional births? If there were no differences in responses, this should be noted.

The findings are contextualized appropriately within the broader literature on home birth practices and barriers to institutional deliveries. Some review of the literature on provider perspectives may further illuminate why women prefer home births instead of institutional deliveries, and how providers view these populations and the services that they deliver.

Are there any illustrative quotes demonstrating mistrust or perceived low quality of care by women toward health center staff? This is a key point in the discussion (lines 418-426) yet the data presented in the results does not sufficiently support this.

Regarding the section on bravery, do the authors have recommendations on how to leverage this information to promote institutional births? What is it about home births that signifies bravery and would it be possible to recreate this within a healthcare setting? This may be beyond the scope of this paper.

Are there any studies related to partner influence on birth decisions that can be referenced to further support the authors’ discussion of partner decision-making and gendered factors?

In making recommendations for clean home birth practices, authors should acknowledge the importance of structural factors such as access to water and sanitation as necessary resources for implementation of hygienic birth practices.

Finally, please proofread for grammatical mistakes throughout.

b. Minor Issues

Line 35: I recommend revising “birth represents a meaningful…practice” to more specific phrasing (e.g., home birth represents a courageous act.”)

Line 81: Please briefly explain how cultural birthing practices differ in the Peruvian Andes as compared to the Amazon, and why cultural integration policies implemented by the Peruvian government have been unsuccessful in the Peruvian Amazon.

Lines 87-88: Who would be the target of clean birth and newborn care interventions – mothers, relatives, traditional birth attendants?

Lines 127-131: How might the exclusion of health center staff perspectives impacted study findings?

Lines 325, 328, & 329: The word “gloves” is misspelled.

Lines 416-417: Can you cite evidence to support this claim?

Line 442: The authors describe the role of the godmother as a “barrier for institutional birth.” Is this cultural practice a barrier in itself or, is the barrier the health center’s reluctance to integrate cultural practices into their delivery services? I recommend reconsidering the language used in this sentence.

Line 499: Do you mean primary care services? Or obstetric services?

Reviewer #2: This is a well-crafted research study on the home birth preference and practices among women in the rural Peruvian Amazon. Due to the general lack of research in this area and the critical need to include indigenous people's voices in questions surrounding their own health care, the qualitative methodology was an appropriate research strategy. The authors clearly described their use of standard approaches for recruiting and engaging participants, for gathering data (interviews), and for their analysis. The results revealed important insights into the barriers that prevent rural Peruvian women and their families from obtaining care in institutional facilities as opposed to their homes, and the discussion contained useful recommendations for community- and institutionally-based care providers. For revision, I would suggest: 1) include a table with descriptive characteristics of the sample, as it was heterogeneous and could be made clearer in table form; 2) include a bit more comparison/contrast from the different participant subgroups (e.g., women who recently gave birth, godmothers, fathers) as the sample heterogeneity raised questions about possible differences; 3) consider discussing how the impact of your efforts could be extended by future research that included health care officials' responses to these and related study findings.

---

## [Author Response · Author response to Decision Letter 0]

12 Feb 2021

We are again grateful to the Reviewers and the Editor of PLoS One for their suggestions, which we are confident have strengthened our paper. Below, we detail the changes we made in response to the comments we received from each reviewer.

Editor:

- Line 29 . Is “every second” literal?

No, it is a figure of speech. We replaced it.

- Line 40. How are drying and cord cutting related here? Again on line 443, the question arises again. Delayed cord clamping is actually recommended to increase iron to the baby. Couldn’t the cord be dried even if there was a delay in having it cut?

By drying we refer to the drying of the newborn, not the drying of the umbilical cord. Drying the baby and cord-cutting are related because the drying happens after the cord is cut. The cutting is done by a godmother who is not always in the room. Then, waiting for her to arrive delays both the cord-cutting and the drying of the newborn.

- Line 41. what are “heterogenous” answers?

We meant that there was a range of responses regarding immediate breastfeeding and immediate drying of the bay. We made changes throughout the paper to clarify this.

- What is new? Is this an update from a previous study? Is there currently a growing stigma for home births? What do people understand about the risk of maternal mortality?

Thank you for your questions, they help us make the Introduction much stronger. 

This is not an update from a previous study, instead, it was preliminary research for a current community-based intervention that trains community health workers to identify pregnancy early, perform home visits to pregnant women and newborns, and promote essential newborn care practices in case institutional birth is not desired or feasible. This study could also contribute to the cultural adaptation of institutional birthing services in the Peruvian Amazon. What is new is that, despite a growing interest in providing culturally appropriated institutional birthing services, home birth and neonatal mortality are still very high in the Peruvian Amazon. We argue that better information about women’s home birth preferences and their childbirth and newborn care practices in the Peruvian Amazon could contribute to both community-based interventions aimed at improving clean birth and newborn practices and the cultural adaptation of institutional birthing services. As a result, the risk of neonatal mortality could decrease. We have re-structured the introduction to make these arguments more clear. 

Perceptions about the risk of maternal mortality were not part of the study. We focused on what made women want to stay at home more than their fears of giving birth at home.

- Line 55 - is this a rate or a percentage?

It is a percentage. We decided to reframe the sentence after re-reading the source. The new version is: “Infections are the main cause of neonatal death in this region, even among newborns with a healthy weight for gestational age and without genetic conditions that might compromise their life [2]”.

- Line 74- more details needed on this integration. When was this? How many facilities?

We included this information in the new version. The first standing birth protocol was issued in 2005 by the Ministry of Health. In 2016 a more detailed protocol was approved. These measures apply nationwide but there is no evidence of its actual implementation. We removed the study from the Andes because it was a community-based intervention run by a non-governmental organization instead of the State and it was conducted before the national standing birth protocols were issued. 

- Line 102 – accessible to where?

We clarified: “Indigenous communities in this geographic area are widely dispersed; their access to health facilities is time-consuming and by boat only”.

- Line 113- were observational data included?

No, only interview data was included.

- Line 126- please include numbers of respondents and justification for the sample size

“Our sample size is 60 participants and was the result of data saturation. The target population consisted of: (1) recent mothers who had experienced childbirth within one year prior to the interview (n=19), (2) female relatives of recent mothers who had a role in childbirth (n=12), (3) male relatives of recent mothers (n=7), (4) community health workers (n=12), and (5) traditional healers (n=10)”.

When we revised the data to provide the number of respondents for each group and create the table with demographic characteristics of our participants we realized we miscounted before. The number of participants is 60 instead of 59.

- Line 130 – spelling should be midwives (or can keep as parteras?); are sobedoras different than TBAs?

We changed traditional midwives to traditional birth attendants throughout the paper. Some sobadoras are TBAs but this is a specific social role so we preferred to make the distinction.

- Line 141- please include consent process details

“These participants were approached by the team at their houses where they received an explanation of the purpose of the study, the risks and benefits of participation and who to contact in case they had questions or concerns”. 

- Line 143- please include the languages, and if tools were pre-tested

“Interviews were conducted in Spanish by R.P. who was supported by Y.S. taking notes. This method was designed based on previous research with the targeted population [6-10] and was not pre-tested”.

- Line 180- why was the stay in each community so short? This would be difficult for a phenomenological study

Budget constraints and difficult living conditions due to limited resources in each village (e.g. lack of water, sanitation, and electricity) made a longer stay impossible. We have realized that our study does not comply with all aspects of the phenomenological approach. We have limited our study design to an interview-based case study.

- Line 192- what are the roles and training of CHWs?

We included the following sentences: “Community health workers are trained to monitor new pregnancies. Their role is to conduct home visits to pregnant women and newborns to check on any signs of health complications and report them to the health facility”.

- If available, please include a demographic table or description of the participants.

This was also suggested by Reviewer 2. In response to both of them, we included Table 1. We included three variables that had a good rate of response (less than 10 missing cases): sex, age, and number of children. We had to exclude years of formal education because almost one third of participants did not answer to that question. This probably due to the access barriers to schools that that affects many communities living in rural areas in Peru.

- Please append the interview guides

They have been included. 

- Line 222- were experiences of mistreatment expressed directly or second hand stories?

Community health workers and recent mothers and their female relatives shared second hand and direct stories, respectively, of women being told off for not pushing during labor and receiving unnecessary procedures such as C-sections.

- Line 266- please spell out SJS

SJS is the abbreviation of San Jose de Samiria, one of the three villages where we conducted our study. To maintain our participants' confidentiality, we have decided to not mention the names of the villages and delete the abbreviations we had in the paper. We also did not find salient differences between the villages so that referring to them does not change the analysis.

- Line 279- is it only gendered factors, or also intrafamiliar, and other power dynamics?

In this subsection, we have focused on gendered factors that happen at both an intrafamilial and an institutional level. In the first case, they are clearly the result of imbalanced power dynamics between men and women. Another power dynamic that we have explored in the paper is the doctor-patient interaction which is marked by mistreatment. There are other intrafamilial factors related to home birth that are not specifically gendered that we mention in other subsections. The feeling of safety and comfort that comes from being surrounded by family members and receiving their help during birth is one example.

- Line 281- what is “L”?

L is the initial of one of the three villages where we conducted our study. To maintain our participants' confidentiality, we have decided to not mention the names of the villages and delete the abbreviations we had in the paper. We also did not find salient differences between the villages so that referring to them does not change the analysis

- Line 313- what types of “things” are asked for?

They are asked for identification, paperwork, and to wait and give birth without relatives in the room. We clarified this in the quote. 

- Lines 325, 328, & 329: The word “gloves” is misspelled.

Thank you, we corrected all the misspelling. We deleted the paragraph on gloves use because it is not part of the best practices for clean births. They are also not distributed to traditional birth attendants by the government and are barely used.

- Line 329- are TBAs supposed to get gloves from health facilities?

No, they are not supposed to get them from the health facilities. In fact, the WHO does not include the use of gloves as part of the procedures for a clean home birth. Because of that, we have decided to remove his part.

- Line 351- were responses not clear, or just given in a range?

They were given in a range. We clarified this in the new manuscript.

- Line 362- is “carried her” the same as skin to skin care?

No, it is not skin-to-skin care because the baby has clothes.

- Line 374- who is “they”?

Professional midwives. We clarified the quote.

- It would be helpful, in the results, to understand the weight of various factors that participants describe that help them decide when and where to seek care.

Thanks for the suggestion, we included this information in the first paragraph of the Home birth preferences subsection: “Study participants reported that most women in their communities prefer to give birth at home instead of a health facility. Feeling safe and supported are the main reasons to stay at home for childbirth. This is contrasted with a fear of mistreatment and inadequate care at health centers. Other related factors include home birth being perceived as brave and partners’ and social expectations that support home birth. Access barriers such as financial and transportation constraints and long wait times are other complementary factors that limit the use of birthing services.

- Line 394- “people want to” should be changed to indicate less choice, given constraints

We replaced that phrase for the following: “In relation to access, studies in the Peruvian Amazon and other low and middle-income countries have also found that distance and financial constraints limit people’s options to home birth [10,23–26].”

- Line 400- lack of awareness by who?

By health providers. We clarified in the manuscript.

- Line 410- who is “their”?

Saker and colleagues. We clarified in the manuscript.

- Line 416- please see the following publication, showing a large percent of women experience unconsented vaginal exams and other disrespectful care 

Thank you, we included the results of the suggested paper in the Discussion section.

- Line 423- what language is spoken at the health care center?

Spanish. We clarified in the new manuscript.

- Line 449- this is not clear – what is the link between pushing and having relatives present? (are the fathers of the babies allowed in health facilities?)

We changed the sentences and re-located them in the paragraph where we discuss safety and support. This is the new version: “Qualitative studies of the Kukama-Kukamiria people show that the presence of relatives during childbirth is understood to drive away bad spirits [5]. The baby’s father also has an important role during childbirth because he motivates the mother when she is pushing [7,35,36]. According to the current standing birth protocol in Peru [14], pregnant women can choose one companion during institutional birth. This can be the father or any other family or community member. In light of our findings and related literature, this norm is insufficient. To make institutional birth more appealing to Kukama-Kukamiria women, more companions should be allowed in the childbirth room”.

- Line 465- are other liquids given when breastfeeding is delayed?

No

- Line 466- who is this reported by?

Moore ER, Bergman N, Anderson GC, Medley N. Early skin-to-skin contact for mothers and their healthy newborn infants. Cochrane Database Syst Rev. 2016 Nov 25;11(11):CD003519. doi: 10.1002/14651858.CD003519.pub4. PMID: 27885658; PMCID: PMC6464366.

- Line 473- please see recent evidence related to the risk of bed sharing - https://pubmed.ncbi.nlm.nih.gov/27940805/

We appreciate this contribution. After reading the paper, we removed the sentences on bed-sharing since it is not an alternative to skin-to-skin contact and it might even be detrimental for the baby. Instead, we added some recommendations to improve skin-to-skin care. " The lack of skin-to-skin contact that we found should be addressed by health officers. We recommend educational campaigns targeting community health workers, pregnant women, and traditional birth attendants about the importance of skin-to-skin contact. Moreover, we recommend addressing one of the main reasons for the delay of such contact – waiting for the godparent to cut the cord. For instance, recommending that the newborn rest on the mother’s chest even if the cord has not yet been cut, is a possibility". When we revised the data to provide the number of respondents for each group and create the table with demographic characteristics of our participants we realized we miscounted before. The number of participants is 60 instead of 59.

- Line 479- the average number of days when colostrum is available is known

Yes, we deleted the sentence. 

- Line 484 – is this maternal or neonatal mortality?

Neonatal mortality. We clarified this in the new manuscript.

- Line 486- use word like “beneficial” rather than suitable, which is more judgmental

Thanks for the suggestion. We changed the word.

- Line 500 – please provide some examples of cultural adaptations

We edited the paragraph. These are some of the examples: “This study shows the limitations of the current standing birth protocol and can inform future changes to ensure the cultural adaptation of birthing services in the Peruvian Amazon. For instance, allowing only one companion might be a barrier to institutional births since women are used to having their close relatives and a birth attendant during childbirth. We recommend increasing the number of companions to at least four people so that the partner, the mother, a childbirth attendant, and a godparent can be present. The cultural adaptation of birthing services should also integrate the role of the godparent in cutting the umbilical cord. The mechanisms to report and hold health providers accountable for mistreatment should also be improved and including community health workers in this process is key”.

- The discussion would be strengthened with more references to the other work that has been conducted on maternal and postnatal care and care seeking in Latin America. Please see the following citations.

We included the suggested papers in the Discussion, thank you for bringing these papers to our attention.

- Please include a strengths and limitations section.

We included a Strengths and Limitations section.

We carefully reviewed the requirements and made the necessary changes. 

- Please include a copy of the interview guide used in the study, in both the original language and English, as Supporting Information, or include a citation if it has been published previously.

We have included the interview guides as supportive materials.

- We note that you have indicated that data from this study are available upon request. PLOS only allows data to be available upon request if there are legal or ethical restrictions on sharing data publicly.

We made data available. 

We made data available. 

- Please amend either the title on the online submission form (via Edit Submission) or the title in the manuscript so that they are identical.

They are identical now, thank you. 

Reviewer 1:

- Line 35: I recommend revising “birth represents a meaningful…practice” to more specific phrasing (e.g., home birth represents a courageous act.”)

Changed to specific phrasing suggested by Reviewer 1.

- It is unclear whether the authors are ultimately recommending improving home birth practices (as is intended with the community-based intervention) or integrating cultural practices into hospital systems to increase institutional births. Even so, health systems must first address the structural barriers impacting access to institutional deliveries, namely distance from facility and cost of transportation. The authors should more clearly indicate what the ultimate implications of their findings are.

Thanks for this input. The primary implications of this study were to inform a community-based intervention aimed at reducing neonatal and maternal mortality through the training of community health workers. They were trained to identify pregnancy early, perform home visits to pregnant women and newborns, and promote essential newborn care practices in case institutional birth is not desired or feasible. This study could also contribute to the cultural adaptation of institutional birthing services in the Peruvian Amazon. This is mentioned in the last sentence of the introduction and the discussion.

- Line 81: Please briefly explain how cultural birthing practices differ in the Peruvian Andes as compared to the Amazon, and why cultural integration policies implemented by the Peruvian government have been unsuccessful in the Peruvian Amazon.

We removed this part from the article because we found no clear evidence that the cultural birthing practices differ between the Peruvian Andes and the Amazon. Moreover, the evidence we presented of the cultural integration in the Andes was a small intervention project conducted by a non-governmental organization previous to Peru's first standing birth protocol. It was not part of a national policy. We apologize for the confusion.

- Lines 87-88: Who would be the target of clean birth and newborn care interventions – mothers, relatives, traditional birth attendants?

The paper cited does not specify who would be the target of clean and birth newborn in the community. But we did include more information about the strategies recommended by the authors. We added these sentences: According to the authors, clean birth and newborn care interventions should be promoted through community-based behavioral change interventions, training of health workers and birth attendants, and clean birth kits [16]. It is unclear who the specific targets of these strategies are except for health workers and birth attendants”.

- Authors should state whether the interviews were conducted in Spanish and if so, how they were translated.

We clarified in the new manuscript that interviews were conducted in Spanish and only the most representative quotes of each theme were translated into English by I.D.M who is bilingual. The entire manuscript has been proofread by a professional English-language editor.

- I am concerned about how conducting the interviews in participants’ homes may have impacted responses, especially in relation to discussions of partners’ decision-making and gender roles in relation to institutional deliveries as participants may have felt uneasy with speaking openly if their partners were present or nearby

We had the same concern about the same so we conducted the interviews with women and men separately. Male relatives were not present or nearby during the interviews with recent women and female relatives.

- Please elaborate on the data analysis methodology – were all transcripts (besides the first two that were double-coded) coded by I.D.M and P.T.? What do the authors mean by “we collectively decided how to organize the themes of this paper” in line 168?

We edited the text to make it more clear: “Authors I.D.M. and P.T. shared the coding book and the main themes that had emerged with the rest of the research team. In this meeting, the research team collectively decided to organize the themes for this paper into two parts – home birth preferences and childbirth and newborn care practices – and to focus the rest of the coding on those sections of the interview guide. Other sections, such as Attitudes and Practices during Pregnancy, Attitudes and Practices in Postnatal and Maternal Care, and Perceptions about Community Health Workers were excluded from the rest of the analysis process because these topics are beyond the scope of this paper”. 

- Lines 127-131: How might the exclusion of health center staff perspectives impacted study findings?

Rather than excluding health center staff, they were not considered as participants because their perspective did not help answer our research question that aimed at understanding home birth preferences and practices around childbirth and newborn care among the Kukama-Kukamiria people. Their participation could have impacted our study findings by providing an outsider perspective on why women prefer home births and an insider experience of institutional births. The latter we could have contrasted with women's experiences and perceptions of institutional births which came out when they described why they prefer home births.

- The results are comprehensive and the quotes illustrative of the identified themes. One way to improve this section would be to tease apart discrepancies between ideas from different types of respondents – what were the differences in perspectives from mothers, partners, traditional healers, community health workers, and female relatives. For example, did postpartum women mention cost or distance as deterrents to institutional births? If there were no differences in responses, this should be noted.

Reviewer 2 had the same suggestion. In response to both of them, we included more contrast between responses, even if they were not differences between types of respondents. For example: "The perception of women who go to health centers as weak or “dejadas,” however, was contested by a few participants from each group. These participants considered that during emergencies, giving birth at the health center is not a sign of weakness. For some of them, women giving birth are always brave regardless of where it happens: “Both of them are brave, those [who give birth] at the health post and those [who give birth] at home. Women are always brave when they have to bring a child into the world.” (male relative, father of 13 children).

We also noted more instances where there were no differences. For example: “Despite the male relatives group who reported not knowing about the cleaning practices before childbirth, all participants mentioned that babies are delivered to a blanket on a previously swept wood floor. The blanket is washed either with rain or river water before the birth takes place. There was only one participant, a recent mother, who did not know if the blanket used during her labor was washed before or not because her mom was in charge of that. A piece of plastic was used by a few participants which goes under the blanket or it replaces it”. 

We discussed why we found few differences between groups in the Strengths and Limitations section.

- The findings are contextualized appropriately within the broader literature on home birth practices and barriers to institutional deliveries. Some review of the literature on provider perspectives may further illuminate why women prefer home births instead of institutional deliveries, and how providers view these populations and the services that they deliver.

We included some of the literature on providers’ perspectives in the discussion:

1) “Peruvian providers’ perspectives of mistreatment during institutional births is unknown, but studies in other countries suggest that providers are aware of verbally and physically abusing women themselves and observing their peers doing the same [31]. Some of the drivers of mistreatment and abuse are perceiving women as being difficult, lack of accountability, and stress [31]”. 

2) “The evidence in Nepal suggests that other barriers to skilled birth care include the limited availability of skilled birth attendants and lack of community awareness regarding the importance of birthing services [27]”.

- Are there any illustrative quotes demonstrating mistrust or perceived low quality of care by women toward health center staff? This is a key point in the discussion (lines 418-426) yet the data presented in the results does not sufficiently support this.

We addressed mistreatment in the first subsection of the results, "Safety and Support v. Mistreatment and Restrictions". There we included first and second-handed stories of mistreatment. We included an additional quote and a new paragraph explaining other sources of mistrust. 

We also changed the name of the subsection to "Safety and Support v. Mistreatment and Inadequate Care". Your comment helped us realize that Inadequate Care is a more adequate description for the data we discussed there. 

- Regarding the section on bravery, do the authors have recommendations on how to leverage this information to promote institutional births? What is it about home births that signifies bravery and would it be possible to recreate this within a healthcare setting? This may be beyond the scope of this paper.

We added some recommendations: “Our study not only reinforces these findings but also shows how the association between home birth and bravery can be a barrier to institutional birth. However, the few people we found who also consider women having institutional births to be brave, suggests a potential cultural change that could lead to more institutional births. To reinforce this cultural change and given what we know about the link between home birth and bravery and the perception of the health center as a place where women are powerless (i.e. subjects of mistreatment and unconsented procedures), we suggest that interventions should emphasize that institutional births are for all women, not just those experiencing complications, and that women can decide their childbirth positions during institutional births. We also recommend allowing more companions than the one that is currently permitted during institutional births”.

- Are there any studies related to partner influence on birth decisions that can be referenced to further support the authors’ discussion of partner decision-making and gendered factors?

We added information from a systematic and a literature review. “In Saker and colleagues’ study [25], the decision to have a home birth was made by the men and elderly members of the family. A systematic review of access barriers to obstetric care at health facilities and a literature review of determinants of institutional births show that women’s decisions are heavily influenced by their partners’ preferences [38,39]. According to these studies, women need their husbands’ or relatives’ permission to have an institutional birth, and they may lack control over the material resources necessary to afford or access one [38,39]”.

- In making recommendations for clean home birth practices, authors should acknowledge the importance of structural factors such as access to water and sanitation as necessary resources for implementation of hygienic birth practices.

Thank you for noting that. We included it in the discussion: “Our study shows that hygienic birth practices (e.g. handwashing, scissors sanitization) are common in the community, but structural factors such as access to water and sanitation are necessary to ensure their effectiveness”.

- Lines 416-417: Can you cite evidence to support this claim?

Although this was stated in one of our references (Anderson, 1999), we decided to eliminate this sentence. We consider that an ethnographic work such as the one by Anderson cannot determine an association between feeling ashamed and maternal death. It is possible that people avoid giving birth in the health center because of those feelings but having birth a home does not necessarily lead to maternal death. Instead, we discussed how home births can be safe for the newborn and the mother if they are clean.

- Line 442: The authors describe the role of the godmother as a “barrier for institutional birth.” Is this cultural practice a barrier in itself or, is the barrier the health center’s reluctance to integrate cultural practices into their delivery services? I recommend reconsidering the language used in this sentence.

Thank you for pointing this out. We edited the language: “It also contributes to it by showing that not integrating the role of the godparents into birthing services can be a barrier to institutional birth”.

- Line 499: Do you mean primary care services? Or obstetric services?

Obstetric services.

Reviewer 2:

- Consider discussing how the impact of your efforts could be extended by future research that included health care officials' responses to these and related study findings.

This was great advice, thank you. We included recommendations in the discussion sections. These are some of those:

1) “Moreover, the implementation of the protocol is limited [15]. In our complementary formative research for the Mamas del Rio program, we found that none of the 10 health facilities serving Kukama-Kukamiria people had the 13 supplies (e.g. mats, ropes, stool) required to adapt their services to accommodate the standing birth protocol. Five of those facilities had only two supplies and the other five had none. Since the lack of implementation of such protocols is a barrier to institutional births, a nationwide evaluation of protocol implementation should be conducted. We recommend that the supplies be distributed to every health center, prioritizing those that primarily serve indigenous people, and that communities be informed about the cultural adaptation of birthing services”. 

2) “We recommend increasing the number of companions to at least four people so that the partner, the mother, a childbirth attendant, and a godparent can be present. The cultural adaptation of birthing services should also integrate the role of the godparent in cutting the umbilical cord. The mechanisms to report and hold health providers accountable for mistreatment should also be improved and including community health workers in this process is key”.

---

## [Decision Letter · Decision Letter 1]

13 Apr 2021

Home birth preference, childbirth, and newborn care practices in rural Peruvian Amazon

PONE-D-20-27331R1

Dear Dr. Del Mastro Naccarato,

We’re pleased to inform you that your manuscript has been judged scientifically suitable for publication and will be formally accepted for publication once it meets all outstanding technical requirements.

Kind regards,

Tanya Doherty, PhD

Academic Editor

PLOS ONE

Additional Editor Comments (optional):

Reviewers' comments:

Reviewer's Responses to Questions

**Comments to the Author**

1. If the authors have adequately addressed your comments raised in a previous round of review and you feel that this manuscript is now acceptable for publication, you may indicate that here to bypass the “Comments to the Author” section, enter your conflict of interest statement in the “Confidential to Editor” section, and submit your "Accept" recommendation.

Reviewer #1: All comments have been addressed

Reviewer #2: All comments have been addressed

2. Is the manuscript technically sound, and do the data support the conclusions?

Reviewer #1: Yes

Reviewer #2: Yes

3. Has the statistical analysis been performed appropriately and rigorously? 

Reviewer #1: N/A

Reviewer #2: Yes

4. Have the authors made all data underlying the findings in their manuscript fully available?

Reviewer #1: Yes

Reviewer #2: Yes

5. Is the manuscript presented in an intelligible fashion and written in standard English?

Reviewer #1: Yes

Reviewer #2: Yes

6. Review Comments to the Author

Reviewer #1: The authors have adequately addressed comments and revised the manuscript accordingly. I recommend this article for publication and believe it will make a significant contribution to the field of research on home births among indigenous populations in Latin America.

Reviewer #2: I am just filling this in because the system will not allow me to submit my review unless I meet a minimum number of characters to submit it -- I am not sure what the problem is.

7. PLOS authors have the option to publish the peer review history of their article (what does this mean?). If published, this will include your full peer review and any attached files.

Reviewer #1: No

Reviewer #2: **Yes: **Elizabeth Soliday

---

## [Editor Report · Acceptance letter]

21 Apr 2021

PONE-D-20-27331R1 

Home birth preference, childbirth, and newborn care practices in rural Peruvian Amazon 

Dear Dr. Del Mastro N.:

I'm pleased to inform you that your manuscript has been deemed suitable for publication in PLOS ONE. Congratulations! Your manuscript is now with our production department. 

Kind regards, 

on behalf of

Professor Tanya Doherty 

Academic Editor

PLOS ONE